# Role of Nutrients in Pediatric Non-Dialysis Chronic Kidney Disease: From Pathogenesis to Correct Supplementation

**DOI:** 10.3390/biomedicines12040911

**Published:** 2024-04-19

**Authors:** Flavia Padoan, Matteo Guarnaroli, Milena Brugnara, Giorgio Piacentini, Angelo Pietrobelli, Luca Pecoraro

**Affiliations:** Pediatric Unit, Department of Surgical Sciences, Dentistry, Gynecology and Pediatrics, University of Verona, 37126 Verona, Italyangelo.pietrobelli@univr.it (A.P.);

**Keywords:** chronic kidney disease, trace elements, nutrition management, supplementation, children’s growth

## Abstract

Nutrition management is fundamental for children with chronic kidney disease (CKD). Fluid balance and low-protein and low-sodium diets are the more stressed fields from a nutritional point of view. At the same time, the role of micronutrients is often underestimated. Starting from the causes that could lead to potential micronutrient deficiencies in these patients, this review considers all micronutrients that could be administered in CKD to improve the prognosis of this disease.

## 1. Introduction

Chronic kidney disease (CKD) is a substantial public health issue, affecting a considerable number of children globally, with prevalence rates ranging from 15 to 74.7 cases per million [1]. This state is characterized by kidney impairment or a glomerular filtration rate (eGFR) below 60 mL/min/1.73 m^2^ persisting for three months or longer, regardless of the underlying cause [2,3]. Ensuring optimal nutrition is crucial for children at every stage of CKD [4]. Proper nutrition is vital to children’s growth and cognitive development [5]. Children with CKD need specialized nutritional attention due to the impairment of their kidney function, hindering the absorption or excretion of certain essential nutrients necessary for achieving typical growth and metabolic equilibrium [6,7]. The International Society of Renal Nutrition and Metabolism (ISRNM) has outlined a concept known as protein–energy wasting. This condition goes beyond reduced food intake, as seen in protein–energy malnutrition. Instead, it characterizes a state in which CKD patients experience decreased body reserves of protein (muscle mass) and energy (body fat), reflecting both nutritional imbalances and wasting processes associated with CKD [8]. This condition involves a complex mix of factors, including hormonal imbalances, decreased appetite, inflammation, increased breakdown of body tissues, loss of nutrients during dialysis and metabolic problems [9,10,11,12]. These combined elements increase the risk of malnutrition, encompassing both undernourishment and overweight, in children with CKD. Changes in body weight, including both weight loss and obesity, have been linked to a faster progression towards end-stage CKD [13,14,15], increased mortality rates among children on dialysis [16,17,18,19] and worse kidney transplant outcome [20,21,22]. Moreover, individuals who are overweight or obese face a higher risk of cardiovascular complications [23,24] and metabolic issues [25,26]. The growth parameters of children with CKD require monitoring at least twice as often as those of healthy children of the same age, with timing adjusted according to the disease stage [27]. Nutrition management is fundamental for children with CKD. In general practice, fluid balance and low-protein and low-sodium diets are the more stressed fields from a nutritional point of view. Monitoring fluid intake in patients with CKD, particularly those undergoing dialysis, is crucial due to the potential cardiovascular complications resulting from intravascular fluid imbalance, which can directly increase the risks of morbidity and mortality [28,29]. While it is essential to avoid fluid overload, excessively restricting fluids may harm the myocardium in pediatric patients [30,31]. Some studies have suggested that choosing between isotonic and hypotonic intravenous fluids does not significantly affect patient management [32,33]. However, evidence indicates that isotonic fluids might offer advantages by reducing the risk of hyponatremia [34]. It is essential to recognize that the sodium requirements for individuals with CKD depend on the root cause of their condition—whether it involves sodium excretion or retention—and their urinary output. Careful monitoring and adjustment of protein intake are essential components of managing the nutritional needs of children with CKD because they play a crucial role in determining mortality rates. Achieving proper growth in these children requires meticulous protein and calorie intake management to maintain a positive nitrogen balance. However, striking a delicate balance is essential, as excessive protein consumption can lead to the buildup of nitrogenous waste products and the onset of uremia, further complicating their condition. Elevated protein consumption might result in heightened intraglomerular pressure and glomerular hyperfiltration potentially contributing to the onset or exacerbation of CKD [35]. The current guidelines of the KDOQI (Kidney Disease Outcomes Quality Initiative) [36,37] advise that children in stages 2–3 of CKD should receive 100–140% of the dietary reference intakes (DRIs) of protein based on their ideal body weight, while those with advanced CKD should aim for 100–120% of the DRIs [38]. Finally, in managing nutrition in children with CKD, the role of micronutrients is often underestimated. CKD heightens the likelihood of vitamin and mineral deficiencies in patients, potentially contributing to the emergence of other health complications like anemia, cardiovascular issues and metabolic disorders. The collective reduction in dietary intake, dietary limitations, impaired absorption in the intestines, inflammation and metabolic acidosis collectively increase CKD patients’ vulnerability to micronutrient deficiencies [39]. This review aims to deepen the role of micronutrients in managing the nutrition of children with non-dialysis CKD. Specifically, we investigate the factors that may contribute to potential deficiencies in these patients and review the current literature regarding the requirements for supplementation.

### 1.1. Sodium (Na)

Sodium (Na) is a mineral that makes up approximately 40% of table salt, while the remaining 60% is chloride. The two terms are generally used interchangeably, which often leads to confusion regarding the correct amount of sodium in the diet (grams of sodium = grams of salt × 0.394) [40]. The American Heart Association (AHA) estimates that about 15% of dietary sodium is naturally present in certain foods, including celery, beets and milk. Only about 11% of total sodium intake comes from what is added while cooking. More than 70% of the sodium we consume comes from processed foods. Sodium flavors food and is used as a binder, stabilizer and food preservative. Sodium or other common forms of sodium (baking soda) are often added to packaged and prepared foods, such as canned soups, deli meats and frozen dinners, which are important sources of hidden sodium. The human body requires a small amount of sodium to conduct nerve impulses, contract and relax muscles and maintain the correct balance of water and minerals. Approximately 500 mg/day of sodium is estimated to be needed for these vital functions. Conversely, too much sodium in the diet can lead to high blood pressure, heart disease and stroke [41]. According to the World Health Organization (WHO), limiting sodium intake to less than 2.3 g/day, corresponding to 5.8 g of salt, is one of the most cost-effective measures for improving public health [42] (Figure 1). Elevated blood pressure values are frequently found in patients with CKD with a consequent increased risk of cardiovascular disease (CVD) [43,44]. Therefore, a low-salt diet (LSD) is central to the treatment of hypertension in CKD. The gold standard for evaluating adherence to LSD is the measurement of sodium excretion by collecting a 24 h urine sample; however, through standardized formulas (Kawasaki, INTERSALT, Tanaka, Nerbass), it is also possible to estimate sodium intake from spot urine samples, which are easier to obtain, at the expense of a greater risk of inaccuracy [45]. Blood pressure monitoring is very important in patients with CKD as hypertension is a frequent complication [46], with a prevalence in this population that progressively increases with the reduction in eGFR. Furthermore, arterial hypertension is the main risk factor for the progression of CKD [47] and cardiovascular mortality [48]. Finally, in patients with CKD, hypertension is often resistant to treatment with a worsening of the cardiovascular prognosis [49,50]. In healthy subjects, high salt intake transiently increases plasma sodium levels, moving water from the intracellular to the extracellular compartment. The increase in plasma sodium levels acts at a central level by stimulating the thirst center and releasing the antidiuretic hormone (ADH) with water reabsorption from the collecting duct. Also, it inhibits the renin–angiotensin–aldosterone system (RAAS) with a reduction of tubular reabsorption of sodium, thus allowing the correct homeostasis of water and sodium to be re-established [51]. In patients with CKD, renal compensation is compromised; therefore, the homeostasis between water and sodium is guaranteed only by the expansion of the extracellular volume, which causes an increase in blood pressure levels. In CKD, the RAAS is inappropriately activated despite expanding the extracellular volume with consequent vasoconstriction of the efferent arteriole and stimulation of sodium reabsorption, contributing to increasing blood pressure levels [52]. Other possible salt deposition sites in the body have also been studied, which, by escaping the direct control of the kidney, can contribute to the release of salt and support a hypertensive state. For example, it has been observed that sodium can accumulate in monocytes and macrophages located in the skin interstitium and can act as osmoreceptors, producing the transcription factor Ton-EBP (tonicity-responsive enhancer-binding protein), which in turn stimulates the production of vascular endothelial growth factor (VEGF), which increases sodium clearance by the lymphatic network [53,54]. Furthermore, high sodium levels in renal failure conditions favor the expression of proinflammatory factors with vascular and cardiac damage [55]. Patients with CKD should limit sodium consumption to improve hypertension control and reduce aberrant RAAS activation with a consequent antiproteinuric effect. The KDOQI guidelines recommend sodium restriction for children with CKD who have hypertension or prehypertension with limits of 1500–2300 mg/day [37] (Table 1). While an LSD diet has been shown to have positive effects in CKD patients not on dialysis, adhering to this diet may be challenging.

### 1.2. Potassium (K)

Potassium is an essential mineral and is the main cation present within cells. It performs numerous functions, including maintaining acid–base balance, allowing the transmission of nervous stimulation and regulating muscular and cardiac activity. Furthermore, potassium is important for the maintenance of cellular osmolarity. In particular, approximately 2% of potassium is found in the extracellular fluid (3.5–5.0 mEq/L) and 98% in the intracellular compartment (140 mEq/L) [56,57], while on the contrary, sodium is more abundant in the extracellular compartment. The diet normally introduces potassium; green fruit and vegetables, beans, nuts, winter squash and dairy products are rich in it. Intake of foods with potassium additives, such as ultra-processed foods and those containing low-sodium salt substitutes or potassium preservatives, is an important hidden source of potassium. Children under one year are estimated to intake approximately 400–700 mg/day of potassium, with a progressive increase of 3000–4500 mg/day for children and adolescents [37]. On average, potassium intake with a Mediterranean diet is 4.8 g/day [58] (Figure 1). Most of the potassium introduced through the diet is absorbed in the small intestine and then distributed into the cells by active transport through the Na-K ATPase. At the renal level, approximately 90% of the potassium taken in through the diet is freely filtered at the glomerulus level, with 70% reabsorbed at the level of the proximal tubule and a further 20% in the thick section of Henle’s loop. In contrast, potassium can be further absorbed or excreted at the level of the collecting duct, depending on potassium levels [59]. Since most ingested potassium is excreted via the kidneys, decreased renal function is an important factor in increasing serum levels, and target values for its intake have been established based on the degree of renal dysfunction. It follows that hyperkalemia is a common CKD complication [60]. Hyperkalemia is a serum or plasma potassium level above the upper normal limits, usually greater than 5.0 mEq/L to 5.5 mEq/L. Mild chronic hyperkalemia is usually asymptomatic or can manifest itself with subtle symptoms such as muscle weakness, nausea and vomiting. Potassium levels above 6.5–7 mEq/L can, however, cause potentially lethal cardiac arrhythmias, muscle weakness or paralysis, especially when the speed of onset is rapid. The main mechanisms involved in potassium homeostasis are renal clearance and the RAAS system, which are gradually impaired in CKD. The kidneys can adapt to the reduced number of nephrons by increasing potassium secretion in residual functioning nephrons, thereby maintaining normokalemia [61]. However, the effectiveness of this adaptation is increasingly hampered as CKD progresses to advanced stages, resulting in the development of hyperkalemia [56,57]. A central point in managing patients with CKD and hyperkalemia is represented by dietary restriction [62]. Although this intervention appears rational, it is supported by limited evidence. It is based on the idea that hyperkalemia in chronic kidney disease is the result of excessive potassium intake, reduced potassium excretion or the redistribution of potassium from intracellular to extracellular space [62,63,64]. A low-potassium diet is generally defined as a dietary potassium intake of 2–3 g/day (50–77 mmol/day) [65], although the exact amount to which dietary potassium should be limited is not defined [63]. The KDOQI guidelines suggest adjusting dietary potassium intake until normal serum potassium values are reached [36] (Table 1). Plant-based diets represent a new perspective on potassium management in patients with CKD. High-potassium fruits and vegetables decrease intestinal transit time, improving fecal potassium excretion and intracellular potassium distribution and potentially helping to reduce extracellular potassium concentrations [62]. Furthermore, plant-based diets provide many antioxidants, vitamins, fibers, flavonoids and carotenoids, improve glycemic control and modulate the intestinal microbiota. Plant potassium has a lower bioavailability (50–60%), and this could allow CKD patients to benefit from these advantages without precipitating hyperkalemia [66,67,68]. In recent years, new potassium-binding agents such as patiromer sorbitex calcium (Patiromer) and sodium zirconium cyclosilicate (SZC) have been developed, providing an important step for chronic CKD management. Patiromer is a non-absorbable polymer that binds potassium in exchange for calcium in the colon. At the same time, SZC is a high-specificity inorganic crystal that delivers potassium into the intestinal tract, allowing the reduction of plasma potassium concentration by increasing fecal excretion of potassium [69]. Further studies are necessary to evaluate whether these drugs can help liberalize the diet in patients with CKD. However, they must be used cautiously to prevent complications such as gastrointestinal side effects (abdominal pain, diarrhea/constipation, dyspepsia, nausea) or interactions with other medications [70]. Another approach to managing potassium in CKD is using loop diuretics, such as Furosemide, which can increase potassium excretion through the urine. However, these medications should be used with caution, especially in patients with compromised kidney function, as they can lead to electrolyte imbalances and worsening kidney function if not monitored closely [71]. Regular monitoring of potassium levels is essential in CKD patients to detect any abnormalities early and to adjust treatment accordingly. This may involve routine blood tests, such as serum potassium levels, and monitoring of renal function and other electrolytes.

### 1.3. Magnesium (Mg)

Magnesium (Mg) is an important mineral that regulates numerous bodily functions. It is an enzymatic cofactor that participates in the synthesis of proteins and DNA. It also plays an energetic and antioxidant role by contributing to glycolysis and glutathione synthesis. Magnesium is important for blood sugar control, blood pressure regulation, bone development, muscle function and nerve impulse transmission, and it contributes to the active transport of calcium and potassium across cell membranes [72,73,74]. The diet represents the main source of magnesium, which is widely contained in green leafy vegetables such as spinach, legumes, nuts, seeds and whole grains [73,74]. In general, foods containing dietary fiber provide magnesium. Magnesium is a primary ingredient in some laxatives [75]. The DRI for children aged 1 to 3 years is 80 mg/day, increasing to 130 mg/day for children aged 4 to 8 years and 240 to 410 mg/day in pre-adolescence and early adulthood [74,76] (Figure 1). Magnesium is mainly absorbed in the intestine via the paracellular route, with bioavailability varying between 20% and 80% [77,78,79], and is also absorbed via active transcellular transport through the magnesium channels TRPM6 and TRPM7 in the distal ileum and colon [77,79]. Finally, a part of magnesium absorption occurs in the jejunum and is influenced by active vitamin D levels, which are often deficient in CKD patients [80]. Magnesium is then excreted at the renal level and 95% reabsorbed at the tubular level [81,82,83]. Magnesium homeostasis depends on various endogenous factors, including parathyroid hormone (PTH), insulin, aldosterone and estrogen levels [77,78]. Most magnesium is present in bones and muscles, while only 1% is present in serum as ionized (70%) or protein-bound (30%) magnesium [77]. Symptomatic magnesium deficiency due to low dietary intake in healthy people is rare because the kidneys limit urinary excretion of this mineral [74]. CKD patients are at risk of developing hypomagnesemia due to dietary potassium restrictions, which can also limit magnesium intake. Furthermore, urinary magnesium excretion is increased by proteinuria and the use of loop and thiazide diuretics. More importantly, tubular dysfunction and interstitial fibrosis reducing renal reabsorption capacity might contribute to magnesium loss [84]. Early signs of magnesium deficiency include loss of appetite, nausea, vomiting, fatigue and weakness. As magnesium deficiency worsens, numbness, tingling, muscle twitching and cramps, seizures, personality changes, abnormal heart rhythms and coronary spasms may occur. Severe magnesium deficiency can cause hypocalcemia or hypokalemia [72]. In the context of CKD, magnesium disorders are associated with increased oxidative stress, production of inflammatory cytokines, sympathetic overactivity, increased adhesion molecules, and inflammation. Indeed, magnesium deficiency has been associated with an increased risk of CVD development and non-fatal and fatal cardiovascular events [85]. Some studies highlight how the relationship between hypomagnesemia and CKD is bidirectional; in fact, magnesium deficiency increases the risk of CKD progression. This is mainly linked to the loss of protection given by magnesium against phosphate-induced renal damage. Sakaguchi et al. studied a population of patients with non-diabetic CKD, and they observed that high serum phosphate levels were associated with an increased risk of progression to end-stage renal disease (ESRD) only when serum magnesium levels were low [86]. High levels of phosphate cause the formation of calcium phosphate crystals, which, accumulating in the proximal tubule, trigger an inflammatory process resulting in tubular damage [87]. Magnesium has a powerful inhibitory capacity for crystallizing calcium phosphate [88], well documented in vascular calcification [89]. In CKD patients, calcium and phosphate bind to fetuin-A to form soluble colloidal particles (CPPs) and prevent nucleation and crystallization. Without such binding, calcium phosphate crystals would trigger an inflammatory process when exposed to macrophages [90]. However, in a high-phosphorus environment, CPPs undergo topological changes from amorphous CPP1 to crystalline CPP2 [91], which can cause oxidative stress with renal damage [87], stimulating the calcification of vascular smooth muscle cells [92]. Magnesium can block the transformation of CPP1 to CPP2 by replacing calcium ions in the structure with a loss of crystallinity [93]. Diaz-Tocados et al. highlighted how a magnesium-rich diet reduces vascular calcification and mortality in uremic rats [94]. Based on these preclinical results, Sakaguchi et al. developed a randomized study involving the administration of oral magnesium oxide to patients with non-dialysis CKD at stages G3 and G4 with evidence of a reduction in the rate of progression of coronary artery calcification in treated patients [95]. Magnesium oxide is a well-known laxative that can reduce the intestinal absorption of phosphate and thus contribute to a reduction in the crystallization of calcium phosphate. The reduction of coronary artery calcification has been associated with a reduction in the risk of CVD in patients with CKD [96]. At the same time, further studies are necessary to evaluate whether there is an actual improvement in prognosis. Hypomagnesemia has also been related to an increased risk of increased blood pressure levels [97], although the mechanisms are not yet fully understood. Magnesium promotes the vasodilation of vascular smooth muscle cells through changes in the activity of the Na+/K+ ATPase, which controls Na+ and K+ transport and levels of the cyclic vasodilators AMP (adenosine monophosphate) and GMP (guanosine monophosphate) [98]. Magnesium also decreases endothelin-1 expression and increases vasodilation by increasing nitric oxide [78]. Finally, aldosterone levels are upregulated in conditions of low magnesium levels, which are reduced following supplementation [99]. Current clinical research has demonstrated that magnesium administration in people with CKD is safe, without concerns for severe hypermagnesemia or negative interference with bone metabolism [100]. However, further studies are needed to define the correct serum magnesium target and which compounds are most appropriate for achieving this target (Table 1).

**Figure 1 biomedicines-12-00911-f001:**
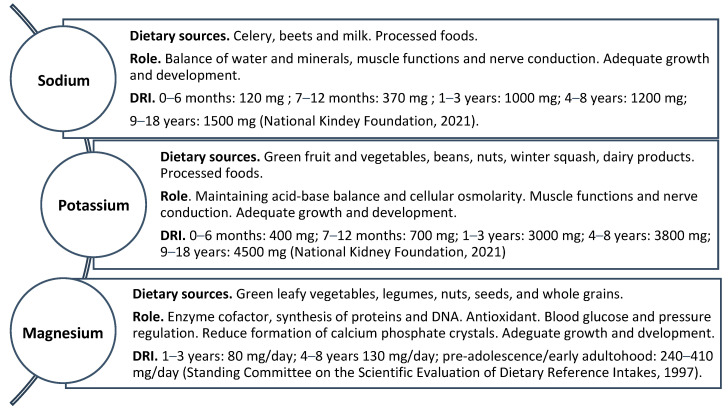
Dietary sources, physiological roles and dietary reference intakes (DRIs) of sodium, potassium and magnesium [37,74].

### 1.4. Calcium (Ca), Phosphate (P) and Vitamin D

Calcium (Ca) is essential in bone health, neuromuscular function, cell signaling, blood clotting and enzyme activation [101]. The main sources of calcium in foods are milk and dairy products; some types of water rich in calcium can also be important sources. Other sources of calcium are legumes, fruit and vegetables, the intestinal absorption of which is reduced due to the presence of fiber, oxalates and phytates [102]. Mineral and bone disorders (MBDs) are prevalent among individuals with CKD, significantly contributing to both morbidity and mortality. CKD-MBDs include biohumoral or laboratory mineral metabolism abnormalities, renal osteodystrophy, cardiovascular disorders and soft tissue calcification [103]. The key organs involved in calcium homeostasis are the gut (absorption), bone (storage) and kidney (excretion), all of which are under the control of calciotropic hormones triggered by the demand for calcium. About 65% of filtered calcium is passively reabsorbed in the proximal tubule and another 25% in the thick ascending limb of Henle’s loop. The remaining 10% is actively (transcellularly) reabsorbed in the distal convoluted tube under PTH control. The concentration of serum calcium in healthy people is maintained within a fairly narrow range from 8.5 to 10.5 mg/dL [104], while the amount of calcium excreted with urine is 4 mg/kg/24 h [105]. In children with CKD, calcium bioavailability is notably impaired, averaging around 15% [106,107,108]. Disturbances in calcium and phosphate homeostasis (vitamin D, calcium, phosphate, FGF23, PTH) heighten the probability of either a negative or positive calcium balance. This condition is due to impaired gastrointestinal function, insufficient vitamin D levels, decreased responsiveness to vitamin D, hypogonadism or dietary restrictions on phosphate intake, which limit the consumption of calcium-rich foods [109,110,111]. Hypocalcemia drives hyperparathyroidism, which, combined with uremia, compromises bone mineralization and increases the risk of fractures in CKD [112]. Calcium demands vary significantly throughout different life stages, peaking during infancy and adolescence [113,114,115]. According to the KDOQI guidelines, it is advised that children receive a calcium intake ranging from 100% to 200% of the age-appropriate DRI, which typically falls between 210 and 1300 mg/day [37], with a maximum limit of 2500 mg/day (Table 1) (Figure 2). This calcium intake should encompass dietary sources rich in calcium, medications like phosphate binders and calcium supplements [36,37].

Phosphate (P) is a constituent element of cell membranes, bones and teeth. Furthermore, it is an enzyme cofactor and pH buffer system. It is important for synthesizing nucleic acids and for the normal function of nerves and muscles [74]. Phosphate in foods is available in two forms: organic (in milk and derivatives, meat, fish, eggs and cereals) and inorganic when added as a preservative in commercial products. Processed foods with significant amounts of added organic phosphate include dairy products, baked goods and meat. Furthermore, inorganic phosphates are added as excipients to various drugs, such as antacids and antihypertensives [102]. Normal serum phosphate levels exhibit significant variations across different age groups within the pediatric population [116]. The DRI of phosphate in children is between 100 and 275 mg/day under 1 year of age and between 460 and 1250 mg/day in children and adolescents [37] (Figure 2). In cases where pediatric CKD patients exhibit serum phosphate levels exceeding the age-appropriate range, dietary phosphate restriction becomes imperative. This necessitates reducing dietary phosphate intake to 80% of the DRI to manage the condition effectively [37] (Table 1).

Vitamin D is essential for skeletal health by contributing to calcium absorption and bone tissue synthesis. Furthermore, it also performs numerous non-skeletal functions, such as metabolic and cardiovascular [117]. Few foods naturally contain vitamin D, but the richest are fatty fish, fish liver oil, egg yolk and cheese [118]. In patients at risk for hypovitaminosis D, the Endocrine Society clinical practice guidelines advocate measuring circulating serum levels of 25-hydroxyvitamin D (25-(OH)D); however, both the Institute of Medicine (IOM) and the Endocrine Society agree that routine screening of the general population is not necessary [119]. The cut-off to define vitamin D deficiency varies based on the guidelines considered; in particular, a value lower than 20 ng/mL is considered deficiency by the Endocrine Society, while it is defined as insufficiency by the IOM, which instead indicates deficiency as values lower than 12 ng/mL [120,121]. Providing infants with supplements to support proper bone development and prevent rickets is crucial. The DRI of vitamin D is 5 μg/day [75] (Figure 2). Exceeding the recommended DRI by 100 times can lead to toxic effects [121]. In pediatric CKD, vitamin D is pivotal in bone health, immune function and overall well-being [122]. Deficiencies stem from multifactorial origins, including poor nutrition, decreased sunlight exposure and urinary losses in proteinuric renal diseases. Additionally, CKD impairs skin vitamin D3 synthesis despite comparable provitamin D3 concentrations [123]. In patients with CKD, particularly in the case of significant proteinuria, hypovitaminosis D is common due to the urinary loss of vitamin D-binding proteins and metabolites [124,125]. Furthermore, in these patients, a low-calcium diet resulting in secondary hyperparathyroidism and increased PTH levels can also lead to low levels of 25-hydroxyvitamin D due to rapid degradation of the same following increased activity of the enzyme 24,25-hydroxylase [126]. In children with chronic renal failure, inadequate levels of vitamin D increase the risk of renal osteodystrophy, characterized by bone abnormalities that worsen with vitamin D deficiency, leading to fractures and growth disorders [127]. Furthermore, vitamin D plays an important role in immune function, which may, therefore, be compromised in patients with CKD by increasing susceptibility to infections [128]. Vitamin D deficiency has also been linked to CVD risk factors, including hypertension, dyslipidemia and endothelial dysfunction. In children with CKD, optimizing vitamin D levels may help mitigate some of these risks. Experimental studies have found a reduction in the incidence of aortic calcification, cardiac/vascular remodeling, left ventricular hypertrophy (LVH), myocardial renin overexpression and blood pressure following the administration of active vitamin D analogues at low doses [129,130]. Given the profound impact of vitamin D on health outcomes in pediatric CKD, routine monitoring and supplementation, as warranted, are integral aspects of patient management (Table 1).

Different therapeutic approaches are employed in managing CKD-MBDs in children, with a primary focus on addressing hyperphosphatemia and secondary hyperparathyroidism. Persistent hyperphosphatemia, despite dietary phosphate restriction, necessitates using phosphate-binding agents. Enteral phosphate binders work by restricting the passive, paracellular absorption of phosphate. They are classified into two primary categories: calcium-based and non-calcium-based phosphate binders. Calcium-based phosphate binders are commonly used and have effectively lowered serum phosphate levels [131]. However, it is important to weigh the benefits of calcium-based binders against the potential risks of hypercalcemia and vascular calcification [132]. Non-calcium-based phosphate binders include Sevelamer formulations, lanthanum and iron-based compounds. Sevelamer is a non-absorbable allylamine hydrochloride polymer that releases chloride upon phosphate binding to cationic amino groups, enhancing metabolic acidosis risk. However, it is very effective in managing hyperphosphatemia without increasing the risk of hypercalcemia [133]. Savelamer is also available as a carbonate, reducing this side effect. Lanthanum carbonate is a trivalent cation that ionically binds phosphate but can accumulate in bone [134]. It is unclear whether this can affect the growth of pediatric CKD patients; therefore, its use is limited in clinical practice. Finally, iron-based phosphate binders can theoretically bind intestinal phosphorus and deliver iron, but their efficacy in pediatric patients has yet to be studied [135].

The primary treatment for secondary hyperparathyroidism is active vitamin D sterols such as calcitriol or its analogues. In pediatric CKD patients in stages 2–4, the initiation of active vitamin D sterols is recommended when serum PTH levels exceed the target range for the respective CKD stage (CKD 2–3 35–70 pg/mL, CKD 4 70–110 pg/mL, CKD 5 non in dialysis > 300 pg/mL), provided that 25-hydroxyvitamin D levels are adequate (>30 ng/mL), corrected total serum calcium is below 10 mg/dL and serum phosphate is within the appropriate range for the patient’s age [131]. Calcitriol, a commonly used active vitamin D sterol, increases the risk of hypercalcemia and hyperphosphatemia. Therefore, vitamin D analogues, such as paricalcitol and doxercalciferol, have been developed with reduced calcemic and phosphatemic effects. Patients with CKD should be re-evaluated frequently with periodic measurements of serum calcium, phosphate and PTH levels. Therapeutic dosage changes should be adjusted using established algorithms [131]. Increasing the dosage of non-calcium phosphate binders in hyperphosphatemia may help reduce phosphate levels by allowing for higher doses of vitamin D sterols. Another class of drugs used to treat hyperphosphatemia and hyperparathyroidism are calcimimetics, including Cinacalcet, which acts by allosterically activating the calcium-sensing receptor (CaSR) [136,137,138]. Finally, in cases of severe and refractory secondary hyperparathyroidism (persistent serum PTH > 1000 pg/mL) in which treatment with vitamin D sterols is ineffective or contraindicated due to persistent hypercalcemia and hyperphosphatemia, subtotal parathyroidectomy may be considered [131].

### 1.5. Iron (Fe)

Iron (Fe) is essential for human health, playing a vital role in numerous physiological processes. The daily iron requirement in pediatrics varies based on age, sex, growth rate and physiological status. For infants aged 6–12 months, the DRI is 11 mg/day, decreasing to 7 mg/day from the second year of life and rising up to 13 mag day in adolescence [139,140,141] (Figure 3). Supplementation is advised only in cases of overt iron deficiency. Excellent dietary sources of heme iron include lean meats, poultry and fish, while legumes, tofu, spinach and other leafy green vegetables contain non-heme iron [142]. Among tissues, bone marrow has a substantial iron requirement for hemoglobin synthesis. Iron deficiency can reduce physical performance due to decreased hemoglobin and myoglobin levels [143]. Iron deficiency can also impair the immune response, reducing macrophage and neutrophil function and decreasing T lymphocytes [144]. Anemia is a common CKD complication primarily caused by iron deficiency. CKD patients may experience anemia due to various factors, including decreased erythropoietin (EPO) production, shortened red blood cell survival, inflammation, hemolysis, blood loss and nutritional deficiencies [145,146]. In patients with CKD, iron deficiency is often due, on the one hand, to an insufficient intake and, on the other, to an excessive loss resulting in anemia due to the reduced production of red blood cells. Furthermore, EPO deficiency or the succession of infectious episodes in these patients can reduce circulating iron levels despite often normal or high iron reserves [147]. Blood iron status is assessed through specific biomarkers, including ferritin, serum iron, total iron-binding capacity and transferrin saturation (TSAT) [148]. New biomarkers are also being studied, including the hepcidin–ferroportin axis, erythroferrone, hypoxia-inducible factors (HIFs), soluble transferrin receptor, hemoglobin content of reticulocytes and percentage of hypochromic red blood cells [149]. However, many of these are affected by renal failure and accompanying inflammation, making it difficult to assess iron status in CKD [150]. The administration of iron to patients with CKD raises concerns due to its essential role in the growth of infectious microorganisms. In treating CKD-related anemia with iron supplementation, there is a risk of microbial proliferation and subsequent infections stemming from excessive iron accumulation. This risk is compounded by heightened hepcidin production, which can hasten iron absorption. Furthermore, the chronic inflammatory state characteristic of CKD may further disrupt iron metabolism, exacerbating the potential for complications [151]. The interplay between iron supplementation, altered iron metabolism and the inflammatory state in CKD can create a complex scenario where microbial proliferation and susceptibility to infections may be heightened. Therefore, careful monitoring and management of iron supplementation are necessary for CKD patients to balance the need for treating anemia with the potential risks of infection. This may involve individualized treatment strategies and close observation of patients for signs of infection. The decision to undertake therapy is based on the patient’s symptoms, laboratory results, health goals and relevant comorbidities. Currently, the first line of treatment for CKD anemia is erythropoietin-stimulating agents (ESAs) combined with iron therapy, which must be adequately metabolized to ensure the effectiveness of ESAs [147]. Iron absorption is compromised in patients with CKD, especially in the more advanced stages, and oral replacement therapy may not be adequate, with the need to consider the intravenous administration route for greater efficacy [152]. Oral administration of iron appears less effective in increasing hemoglobin, ferritin and TSAT serum levels than intravenous injection. Furthermore, gastrointestinal adverse reactions, malabsorption and disturbances of the intestinal flora may occur [153]. Another aspect that influences oral iron absorption in CKD is inflammation, which increases hepcidin and reduces iron absorption in the gastrointestinal tract [154]. However, intravenous iron therapy effectively bypasses the hepcidin pathway, leading to a more rapid replenishment of iron stores [155]. Iron therapy aims to replenish reserves and increase hemoglobin levels to desired levels through adequate red blood cell production. The guidelines recommend that patients with anemic CKD who are not receiving ESA therapy and not receiving hemodialysis (HD) be first offered a trial of oral iron. If these patients are intolerant or the target hemoglobin levels are not achieved within three months, intravenous iron therapy should be implemented [156] (Table 1).

### 1.6. Selenium (Se)

Selenium (Se) is present in multiple proteins as a constituent of various amino acids. Selenium plays a crucial role in safeguarding cells and tissues from oxidative harm and supporting the proper functioning of the immune system. Additionally, it aids in mitigating the toxicity of heavy metals, contributes to the formation of thyroid hormones, facilitates nucleotide synthesis and is essential for growth and reproduction [157]. The concentration of selenium in the serum of healthy people is between 0.5 and 2.5 μmol/L, and, for children, the DRI is between 15 and 55 μg/day [158,159] (Figure 3). Brazil nuts, seafood and organ meats are the richest food sources of selenium. Other sources include muscle meat, grains and dairy products. Selenium is absorbed at the intestinal level, mainly in the cecum and duodenum, and then excreted at the renal level (the daily excretion of selenium is 13.5 ± 3.8 ug/g Cr for women and 9.8 ± 3.3 ug/g Cr for men) [160]. Selenium reserves are found mainly in the kidneys and liver. In patients with CKD, selenium deficiency may be due to poor dietary intake, alteration of intestinal absorption and reduced production of selenium-binding protein; furthermore, part of the selenium may be lost in the urine or during dialysis [161,162,163]. Selenium deficiency increases the risk of atherosclerosis (aggregation of thrombocytes), atopic diseases (leukotriene biosynthesis), inflammatory diseases and infectious diseases (reduced antimicrobial activity of neutrophils, IgG and IgM concentrations in serum, antibody production, proliferation and differentiation of T lymphocytes). Selenium deficiency occurs when the daily DRI intake of the element falls below 40–45 ug. Selenium deficiency has been associated with cardiovascular damage, atopic asthma and infertility in both males and females [164]. The association between selenium levels and kidney function has been debated for years. Traditional observational studies have indicated that individuals with CKD typically have lower plasma and serum selenium levels than those in the general healthy population [165,166,167,168], and such levels tend to decrease with disease progression and eGFR reduction. Ensuring adequate selenium intake could produce beneficial effects on the progression of chronic kidney disease [169,170,171,172]. In contrast, some studies have not identified a connection between selenium levels and CKD [173], and no benefit on metabolic or inflammatory outcomes has been observed following supplementation in patients undergoing HD [174,175]. As part of its role as an antioxidant, the best-studied function of selenium is that of a constituent of the enzyme glutathione peroxidase (GSH-Px) [176] produced by the renal proximal tubule cells. GSH-Px deficiency contributes to the induction of oxidative stress, resulting in the development of an inflammatory state at the renal level with activation of nuclear factor kappa B (NF-kB) and the HIF-1 pathway [177,178]. Selenium can inhibit NF-kB activation by modulating gene expression. Selenium plays an anti-inflammatory role by reducing C-reactive protein (CRP) levels [178] and stimulating the production of immune cells, including T lymphocytes and natural killer cells [179]. Therefore, selenium supplementation could potentially slow down the progression of the disease and renal fibrosis in patients with CKD. However, the data are still controversial. In children with CKD who are conservatively managed, selenium blood concentrations are normal in the majority of cases, indicating that routine monitoring of this trace element is not required in those children [180]. Also, the KDOQI guidelines do not suggest routine selenium supplementation in adults with CKD stage 1-D5 [36] (Table 1).

### 1.7. Zinc (Zn)

Zinc (Zn) plays pivotal roles in cellular and subcellular regulation, boasting antioxidant and anti-inflammatory properties [181,182]. Dietary sources such as dairy products, seafood, meat and poultry primarily contribute to zinc intake, with absorption occurring in the intestine followed by eventual excretion in the stool [183]. The DRI for zinc in infants is 2.9 mg/day, rising to 4.3 mg/day in toddlers. In adolescence, the DRI is 11 mg/day [139,140] (Figure 3). Zinc deficiency represents a significant global micronutrient insufficiency, particularly impacting developing countries and contributing to various health issues [184]. Zinc deficiency has a wide variety of clinical manifestations that include slowed growth, decreased muscle mass, delayed cognitive development, weakened immune function, primary hypogonadism, diminished taste and smell perception and other manifestations related to the degree of deficiency [185,186,187]. In individuals with CKD, circulating zinc levels tend to be lower, while urinary zinc excretion over 24 h is higher, possibly due to faulty tubular reabsorption mechanisms [188]. Zinc supplementation results in a modest but noteworthy improvement in nutritional status among children and adolescents diagnosed with CKD [189]. Some studies have shown that serum zinc concentrations are associated with renal function; in particular, a linear relationship was observed between a reduction in basal zinc levels and a decline in renal function. Zinc concentrations can accurately predict the preservation of renal function regardless of albumin levels, eGFR, age, gender and other associated health conditions [190]. However, in a study by Damianaki et al., the association between serum zinc levels and reduced renal function lost significance once eGFR or baseline proteinuria were included in the analysis [188]. Low zinc levels could represent a risk of progression to ESRD in individuals with advanced CKD [191]. Another important role zinc plays in CKD is a defense against phosphate-triggered calcification; in particular, zinc can reduce the likelihood of calcification of the abdominal aorta and the risk of CVD [192,193]. In CKD, the risk of CVD is also increased by the development of a fibrotic process involving not only the kidneys but also the heart, leading to severe myocardial dysfunction [194,195]. Fibrosis marks the culmination of maladaptive repair, characterized by the proliferation of local mesenchymal cells and the accumulation of the extracellular matrix. The fibrotic process in CKD involves a complex interaction involving inflammation, RAAS, PTH, FGF23/klotho axis, microRNAs and vitamin D [194]. Some studies have highlighted the antifibrotic properties of zinc in the liver and lungs [196,197,198], and it is promising in reducing the risk of fibrosis progression even in patients with chronic renal failure. Current findings indicate that supplementation improves nutritional health and mitigates oxidative stress among individuals with zinc deficiency. However, more evidence is needed to determine whether zinc supplementation improves the overall survival or kidney function in non-dialysis CKD patients. Although the specific zinc requirements of patients with chronic kidney disease remain undetermined, the KDOQI recommends monitoring dietary zinc intake based on the DRI for children and adults, and it may be advisable to evaluate serum zinc levels before considering integration [191] (Table 1). Regular monitoring and supplementation as necessary are recommended, especially for children in developing countries, where baseline DRI levels may be insufficient [189]. Supplementation shows promise in mitigating nutritional deficits and slowing fibrotic processes, but further research is needed to clarify its impact on renal outcomes. Regular monitoring of zinc levels and adherence to dietary recommendations are essential to optimize nutritional intake in patients with chronic renal failure. Integrating zinc management into comprehensive care strategies can improve this population’s health outcomes and quality of life.

### 1.8. Copper (Cu)

Copper (Cu) is the body’s third most abundant trace element [151]. Copper is an important enzymatic cofactor for energy production and iron absorption. It plays a role in producing collagen, connective tissue and brain neurotransmitters. Copper supports normal brain development and immune functions and performs important antioxidant actions. Copper serum concentration is stable, usually between 1 and 1.3 mg/mL of serum [199,200]. Copper is high in protein foods such as organ meats, shellfish, fish, nuts and seeds, whole grains and chocolate [201]. The DRI for copper from foods ranges from 800 to 1000 μg/day for children aged 2 to 19 years [202,203] (Figure 4). After being absorbed at the intestinal level, copper binds to albumin, transporters or low-molecular-weight histidine–copper complexes and is transported to the liver and distributed to other tissues, mainly thanks to the protein ceruloplasmin. Normally, copper flows in an enterohepatic circulation, and only a small part is eliminated in the bile [204].

Children with CKD are at risk of developing copper deficiency due to dietary restrictions, malnutrition and gastrointestinal malabsorption. Another element that can contribute to deficiency in these patients is zinc supplementation, which antagonizes the absorption of divalent cations, including iron and copper, in erythrocyte precursors [205]. Copper deficiency can also cause hematological or neurological symptoms, as this element is important for the functioning of various copper-dependent enzymes [206]. Furthermore, in individuals with copper deficiency, the lifespan of erythrocytes is shorter due to an accelerated oxidation reaction. These factors together make children with chronic renal failure prone to anemia [207]. The main biomarkers used to diagnose copper deficiency anemia are the serum levels of copper, ceruloplasmin and zinc if supplementation is underway (every 2–3 months). Furthermore, the levels of copper in 24 h urine must be measured. Copper deficiency anemia is more common in patients on HD and is treated with oral or intravenous copper replacement with copper gluconate, copper sulfate or copper chloride [207]. Some side effects related to copper supplementation may include nausea and vomiting, likely due to stimulation of the vagus nerve [204]; instead, symptoms of copper intoxication include melena, jaundice, anorexia, severe thirst, liver and kidney failure [208] and copper deposits in the cornea, kidneys and bones as in Wilson’s disease [209]. On the other hand, in the more advanced stages of CKD, an increase in copper levels has been observed [210], which can directly catalyze the formation of reactive oxygen species (ROS) or significantly reduce glutathione levels [211], resulting in oxidative stress and activation of the inflammatory process [212]. In more severe cases, excess copper can cause granular and vacuolar degeneration of renal tubular epithelial cells with the development of renal fibrosis and progression of renal failure [213]; therefore, in CKD patients, it is important to monitor copper levels to prevent both the deficit and the accumulation. Copper supplementation is not suggested in children or adults with conservatively managed CKD [36] (Table 1).

### 1.9. Manganese (Mn)

Manganese (Mn) is another crucial trace element within the body. Manganese is an important metal for human health, necessary for development, glucose metabolism and the antioxidant system [214]. Manganese deficiency is associated with bone demineralization, poor children’s growth, skin rashes and hair depigmentation [140]. On the other hand, excessive accumulation of manganese in various tissues, including the brain, heart and liver, can potentially lead to cognitive impairment, tremors and dystonia, hypotension and cardiac depression [215,216]. Normal blood levels of manganese typically fall in the range of 4–15 μg/L [217]. The DRIs for manganese are 0.003–0.6 mg/day for infants and 1.2–1.5 mg/day for children [139] (Figure 4). Manganese is present in a wide variety of foods, including whole grains, clams, oysters, mussels, nuts, soybeans and other legumes, rice, leafy greens, coffee, tea and many spices such as black pepper [151,218]. The intestinal absorption of manganese is very poor (maximum 10%) and undergoes competition from iron and cobalt. It is then eliminated with the bile, and only a small fraction (0.1–3%) is excreted in the urine [219], thus explaining the lack of significant changes in manganese levels in CKD patients [220]. Studies on manganese levels in patients with CKD are limited and influenced by the patient’s diet, the extent of malnutrition and gastrointestinal malabsorption. As previously mentioned, manganese absorption is influenced by iron; in particular, it increases when the iron content is low, which is common in patients with CKD who present various degrees of iron deficiency [215]. Few pediatric studies have demonstrated manganese deficiencies in dialysis-dependent patients [221,222]. The primary function of manganese is as a cofactor for manganese superoxide dismutase (MnSOD), which is highly active in the human kidney, particularly within the inner mitochondrial membrane [223]. MnSOD dysfunction has been studied in animal models of kidney disease, and it has been associated with increased renal fibrosis, inflammation and renal damage from excessive oxidative stress contributing to the onset of acute kidney injury (AKI), worsening of CKD and the transition from AKI to CKD [223]. Despite the importance of manganese, no guidelines currently recommend daily supplementation of the element in case of deficiency, particularly in children managed conservatively (Table 1).

### 1.10. Iodine (I)

Iodine (I) is a fundamental element in the production process of the thyroid hormones thyroxine (T4) and triiodothyronine (T3), which play an essential role in the regulation of metabolism, growth and development. Thyroid hormones and renal function are related to each other [224]. Thyroid hormones are metabolized at the kidney level. Still, at this site, they influence fluid and electrolyte balance, glomerular filtration rate, renal architecture, renal blood flow and tubular function [225]. Good food sources of iodine include cow’s milk and dairy products, eggs, marine fish, shellfish and algae, although in most countries, the main food source is fortified salt [226,227]. In the fetus and newborn, an adequate supply of iodine is provided by breast milk if the mother consumes at least 250 μg/day in her diet [228]. Iodine deficiency at this stage of life can lead to the onset of hypothyroidism and cretinism with serious cognitive deficits [140,141]. The DRI for iodine is 90–120 μg per day [140,229] (Figure 4). In children with chronic kidney disease, iodine intake may be suboptimal due to dietary restrictions, including limitations on foods rich in potassium, phosphate and sodium, whose blood levels tend to increase with disease progression and reduction of urinary excretion. As a result, accumulation of iodine content occurs in the thyroid gland, resulting in enlargement [151]. Iodine status is assessed by measuring thyrotropin (TSH) and thyroglobulin (TG) levels, as well as urinary iodine concentration (UIC) and thyroid volume. Maintaining optimal iodine levels is important for general health and well-being [230]. In children with chronic renal failure, where metabolism and growth are already compromised, it is very important to maintain adequate iodine levels with good thyroid function. Monitoring iodine status is, therefore, essential to prevent complications related to both a state of iodine deficiency and excess (Table 1).

**Figure 3 biomedicines-12-00911-f003:**
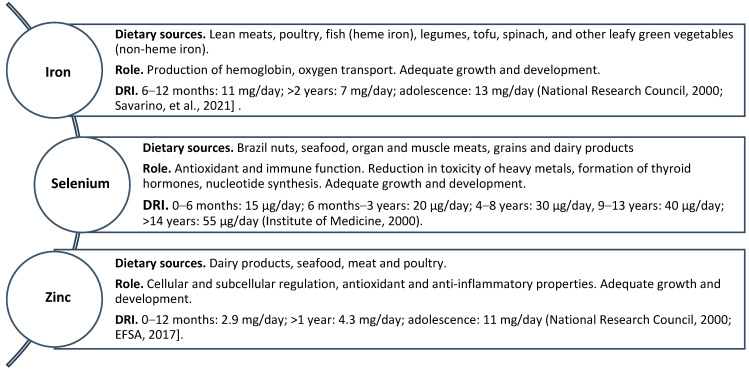
Dietary sources, physiological roles and dietary reference intakes (DRIs) of iron, selenium and zinc [139,140,158,231].

**Figure 4 biomedicines-12-00911-f004:**
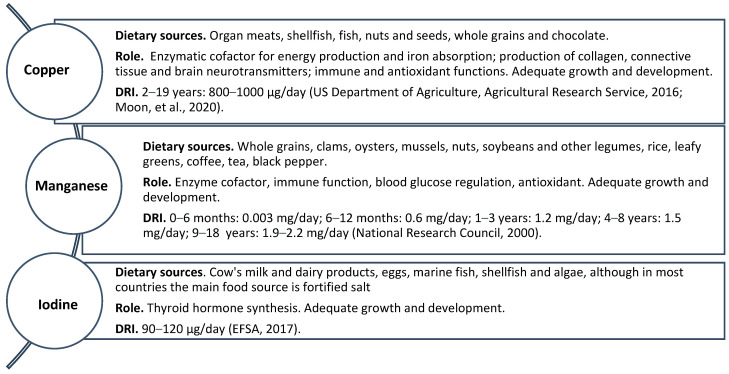
Dietary sources, physiological roles and dietary reference intakes (DRIs) of copper, manganese and iodine [139,140,202,203].

**Table 1 biomedicines-12-00911-t001:** Manifestations of primary trace element alterations in pediatric patients with chronic kidney disease (CKD) and recommendations for management [37,74,232,233].

Micronutrients	Alteration	Signs and Symptoms	Management
Sodium (Na)	Hypernatremia	High blood pressure.	Dietary restriction in case of hypertension or prehypertension. 2–3 years: 1500 mg/day; 4–8 years: 1900 mg/day; 9–13 years: 2200 mg/day; >14 years: 2300 mg/day.
Potassium (K)	Hyperkalemia	Usually asymptomatic. Muscle weakness, nausea, vomiting. Lethal cardiac arrhythmias if rapid onset.	Dietary potassium intake must be less than 2–3 g/day if persistent hyperkalemia uses potassium-binding agents. Adjusting dietary potassium intake until normal serum potassium values are reached.
Magnesium (Mg)	Hypomagnesemia	Loss of appetite, nausea, vomiting, fatigue, weakness. In case of severe magnesium deficiency, numbness, tingling, muscle twitching, cramps, seizures, personality changes, abnormal heart rhythms, hypocalcemia or hypokalemia.	Dietary supplementation with magnesium-enriched products and oral supplements. Further studies are needed to define the correct serum magnesium target and which compounds are most appropriate for achieving this target.
Calcium (Ca)	Hypocalcemia	Muscle cramps and spasm, bone pain, fractures, altered mental state, heart arrhythmia, skin alterations.	Dietary intake is 800 to 1000 mg/day (max 2500 mg/day), including calcium-containing foods, medications such as phosphate binders and calcium supplements to maintain a neutral calcium balance.
Phosphate (P)	Hyperphosphatemia	Secondary to hypocalcemia	Dietary restriction to 80% of daily intake. If hyperphosphatemia persists, use phosphate-binding agents.
Vitamin D	Hypovitaminosis D	Rickets, secondary hyperparathyroidism	If deficiency 2000 UI/day for 6 weeks, then maintenance with 400–800 UI/day.
Iron (Fe)	Iron deficiency	Anemia, fatigue and weakness, headache, brittle nails or hair loss.	In case of anemia, 2–6 mg/kg (max 200 mg) daily of elemental iron in 2–3 divided doses. According to WHO guidelines, 30–60 mg/day in adolescent girls.
Selenium (Se)	Selenium deficiency	Immunodeficiency, fatigue and weakness, infertility, atherosclerosis and atopic disease.	No guidelines currently recommend daily supplementation in deficiency cases, particularly in children managed conservatively. The goal is to optimize nutritional intake.
Zinc (Zn)	Zinc deficiency	Slowed growth, decreased muscle mass, delayed cognitive development, weakened immune function, primary hypogonadism, diminished taste and smell perception.	No guidelines currently recommend daily supplementation in deficiency cases, particularly in children managed conservatively. The goal is to optimize nutritional intake.
Copper (Cu)	Copper deficiency Copper excess in higher stages	Anemia. Neurological symptoms. Degeneration of renal tubular epithelial cells, renal fibrosis and renal failure.	No guidelines currently recommend daily supplementation in cases of deficiency, particularly in children managed conservatively. The goal is optimizing nutritional intake.
Manganese (Mn)	Rare, possible deficiency in dialysis-dependent patients	-	No guidelines currently recommend daily supplementation in cases of deficiency, particularly in children managed conservatively.
Iodine (I)	Iodine excess	Impaired thyroid function, goiter	Dietary restriction. Excessive dietary restriction can cause iodine deficiency.

## 2. Discussion

In pediatric non-dialysis CKD patients, alterations in ions and trace elements play a crucial role in disease progression and overall health. These changes reflect the impaired renal function and disrupted mineral metabolism characteristic of CKD [231]. Electrolyte imbalances, including sodium, potassium, calcium and phosphate level alterations, are commonly observed in non-dialysis pediatric CKD patients. Hypertension is a common complication of chronic kidney disease due to sodium and water imbalance and RAAS dysregulation [52]. The key point in preventing this condition is limiting dietary sodium intake. Similarly, hyperkalemia, a potentially life-threatening condition, results from the impaired excretion of potassium by the kidneys. Management involves dietary restriction of potassium-rich foods and medications to enhance potassium elimination [9]. Conversely, hypocalcemia often arises due to impaired vitamin D metabolism and decreased calcium absorption, leading to secondary hyperparathyroidism and renal osteodystrophy [127]. Phosphate retention is also common, contributing to hyperphosphatemia, which further exacerbates mineral and bone disorders in pediatric CKD. Furthermore, alterations in trace elements such as iron, zinc and copper are prevalent in non-dialysis pediatric chronic kidney disease, contributing to the slow growth and development of these patients and predisposing them to anemia. Anemia in CKD is characterized by decreased EPO production and shortened red blood cell lifespan, necessitating iron supplementation to optimize hematologic parameters and mitigate symptoms. However, iron overload can occur with excessive supplementation, leading to oxidative stress and tissue damage [145]. Careful monitoring of iron status and appropriate dosing are essential to avoid complications. Zinc deficiency is another notable alteration observed in pediatric CKD patients. Reduced intake, impaired absorption and increased losses contribute to zinc deficiency, impairing growth, immune function and wound healing. Supplementation may be necessary to maintain adequate zinc levels, although caution must be exercised to prevent toxicity [191]. Furthermore, alterations in other trace elements, such as copper and selenium, may occur in pediatric CKD [151]. Copper deficiency can manifest as hematologic abnormalities and neurological symptoms [207], while selenium deficiency may impair antioxidant defenses and exacerbate oxidative stress in CKD [165].

## 3. Future Directions

Research into the significance of trace elements in pediatric CKD is essential for exploring various critical domains, thus enhancing comprehension of the relationship between trace element supplementation and patient outcomes. Firstly, there is a need for longitudinal studies assessing the impact of trace element deficiencies on long-term clinical outcomes in pediatric CKD populations. Future goals include investigating the relationship between trace element status and the progression of CKD, as well as the development of cardiovascular complications, bone health issues and other comorbidities. Further research is warranted to elucidate the mechanisms underlying trace element deficiencies in pediatric CKD, including factors contributing to altered metabolism and absorption of these micronutrients. Understanding these mechanisms will help develop targeted therapeutic interventions to address trace element deficiencies effectively. Moreover, studies should focus on evaluating the efficacy and safety of different supplementation strategies for correcting trace element deficiencies in pediatric CKD patients, considering factors such as dosage, formulation and duration of treatment. Furthermore, there is a need for studies investigating the potential interactions between trace element status and other aspects of CKD management, such as pharmacotherapy and dietary interventions. Additionally, given the potential role of trace elements in modulating immune function and oxidative stress, future studies should explore their impact on inflammation and infection risk in pediatric CKD populations. Addressing these research priorities will help inform clinical practice guidelines and optimize the management of trace element deficiencies in pediatric CKD, ultimately improving patient outcomes and quality of life.

## 4. Conclusions

This comprehensive review thoroughly dissects the intricate interplay of essential nutrients for managing non-dialysis CKD pediatric patients. With an unwavering focus on elucidating pathological alterations and the kidney’s role in their imbalance, it covers a spectrum including sodium, potassium, magnesium, calcium, phosphorus, vitamin D, iron, selenium, zinc, copper, manganese, iodine, sodium, potassium and magnesium. Each nutrient’s physiological role, daily intake requirements and potential supplementation in children are explored. This review delves into the multifaceted roles of nutrients in CKD management, starting with sodium, potassium and magnesium. It traverses through their involvement in maintaining electrolyte balance, fluid homeostasis, and cardiovascular function. Emphasizing the precarious balance crucial for CKD patients, the review details the nuanced nutritional strategies necessary to mitigate complications like hypertension and electrolyte disturbances, ultimately aiming to optimize patient outcomes and quality of life. Sodium and potassium dietary restriction is the key point in the management of these conditions, and in children with CKD who have hypertension or prehypertension, dietary sodium intake should be limited to 1500–2300 mg/day while potassium intake should be adjusted until normal serum potassium values are reached. Calcium and phosphorus are highlighted as irreplaceable in preserving skeletal integrity. In disrupted mineral metabolism in CKD, customized supplementation regimens are essential to prevent severe conditions like renal osteodystrophy. Children with CKD should receive a calcium intake ranging from 100% to 200% of the age-appropriate DRI, typically between 210 and 1300 mg/day, while the phosphate intake should be reduced to 80% of the DRI to manage hyperphosphatemia effectively. Vitamin D’s crucial role in calcium absorption, bone health and immune system modulation is underscored. An adequate Vitamin D supplementation with 2000 UI/day for 6 weeks in case of deficiency and then maintenance with 400–800 UI/day is important to ensure proper levels. Iron supplementation is a cornerstone in combating anemia, a prevalent complication in pediatric CKD. The importance of advocating for nuanced supplementation approaches involving meticulous monitoring of iron levels and treatment responses is emphasized to handle this complication and improve patients’ overall well-being effectively. Patients with anemic CKD should first be offered an oral iron trial. If the patients are intolerant to this or target hemoglobin levels are not achieved within three months, intravenous iron therapy should be implemented. Selenium and zinc are critical in mitigating oxidative stress, bolstering immune competence and safeguarding cardiovascular integrity. Tailored supplementation approaches that account for individual patient needs are deemed indispensable. While solid data support the supplementation of sodium, potassium, vitamin D and iron when deficiencies occur in children with CKD, the necessity and extent of supplementation for other nutrients such as copper, manganese and selenium remain under debate. Conflicting opinions on the need and amount of supplementation needed to ensure adequate nutritional intake in children with CKD highlight the need for further studies to precisely define supplementation requirements and optimize the health and well-being of these patients. Healthcare providers are recognized to have a pivotal role in navigating the complexities of micronutrient management in pediatric CKD. A collaborative, multidisciplinary approach integrating nephrology, nutrition and pharmacy insights is necessary to ensure comprehensive and individualized care for each patient.

## Figures and Tables

**Figure 2 biomedicines-12-00911-f002:**
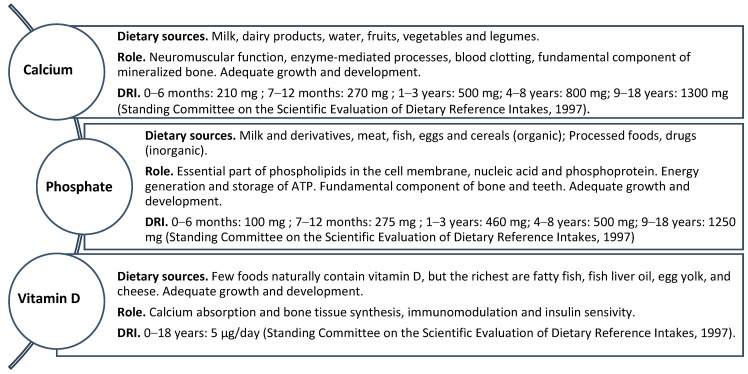
Dietary sources, physiological roles and dietary reference intakes (DRIs) of calcium, phosphate and vitamin D [74].

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
