# Peer review of "Role of Nutrients in Pediatric Non-Dialysis Chronic Kidney Disease: From Pathogenesis to Correct Supplementation"

_biomedicines, 2024, doi:10.3390/biomedicines12040911_

Round 1

Reviewer 1 Report

Comments and Suggestions for Authors

CKD represents a major health issue and it is common knowledge that nutrition is an important part of this disease management, especially in children with renal impairment. Therefore, the present work could highlight significant aspects that should be taken into account in the assessment of this particular group of population. Even if the presented data describes several important nutritional points, for a better understanding, you should include tables with dietary recommendations of these micronutrients (and their potential associated adverse effects). Additionally, it would be nice if you could include some figures (for each of the presented micronutrients) describing the underline mechanisms and their impact on the renal function, and children's growth and neurocognitive development.

Author Response

CKD represents a major health issue and it is common knowledge that nutrition is an important part of this disease management, especially in children with renal impairment. Therefore, the present work could highlight significant aspects that should be taken into account in the assessment of this particular group of population. Even if the presented data describes several important nutritional points, for a better understanding, you should include tables with dietary recommendations of these micronutrients (and their potential associated adverse effects). Additionally, it would be nice if you could include some figures (for each of the presented micronutrients) describing the underline mechanisms and their impact on the renal function, and children's growth and neurocognitive development.

Thanks for the suggestion. We have included 4 figures that help to visualize the dietary sources, the physiological role and the DRI of each nutrient. The figures are referenced in the text.

Reviewer 2 Report

Comments and Suggestions for Authors

The paper provides a comprehensive review of the role of various nutrients in managing the nutrition of children with chronic kidney disease (CKD). It covers essential elements such as calcium, phosphorus, vitamin D, iron, selenium, zinc, copper, manganese, and iodine, highlighting their importance, deficiencies, and potential supplementation strategies in pediatric non-dialysis CKD. However, there are some omissions and areas for improvement in the paper.

The title "Role of Micronutrients in ..." is not entirely accurate because calcium, phosphate, and vitamin D are not classified as micronutrients. This raises the question of why sodium was not included in the paper. Potassium and magnesium also are known to play crucial roles in CKD. Therefore, the authors should either limit the paper to microelements only or include the aforementioned nutrients and adjust the title to something more accurate, such as "Role of Nutrients in ...".

The manuscript suffers from redundancy and an illogical structure. The Introduction section introduces all included nutrients, while immediately following it, the Discussion section appears. Additionally, potassium is mentioned in the Discussion section without being introduced previously. Ideally, the Discussion should focus on analyzing findings, interpreting implications, and exploring avenues for future research. Enhancing this section with more detailed discussions on the efficacy, safety, and optimal dosing of supplementation would improve the paper's quality.

Abbreviations should be defined just once in the text to avoid repetition. For example, the term "CKD" was defined at least 18 times throughout the text.

While the paper briefly mentions potential interactions between nutrient status and other aspects of CKD management, such as pharmacotherapy and dietary interventions, exploration of these interactions and their implications for patient outcomes would strengthen the paper.

Ensure that Table 1 is referenced in the text.

Author Response

The paper provides a comprehensive review of the role of various nutrients in managing the nutrition of children with chronic kidney disease (CKD). It covers essential elements such as calcium, phosphorus, vitamin D, iron, selenium, zinc, copper, manganese, and iodine, highlighting their importance, deficiencies, and potential supplementation strategies in pediatric non-dialysis CKD. However, there are some omissions and areas for improvement in the paper. The title "Role of Micronutrients in ..." is not entirely accurate because calcium, phosphate, and vitamin D are not classified as micronutrients. This raises the question of why sodium was not included in the paper. Potassium and magnesium also are known to play crucial roles in CKD. Therefore, the authors should either limit the paper to microelements only or include the aforementioned nutrients and adjust the title to something more accurate, such as "Role of Nutrients in ...".

Thanks for the clarification; we have added three paragraphs on sodium, potassium and magnesium. We also changed the title.

The manuscript suffers from redundancy and an illogical structure. The Introduction section introduces all included nutrients, while immediately following it, the Discussion section appears. Additionally, potassium is mentioned in the Discussion section without being introduced previously. Ideally, the Discussion should focus on analyzing findings, interpreting implications, and exploring avenues for future research. Enhancing this section with more detailed discussions on the efficacy, safety, and optimal dosing of supplementation would improve the paper's quality.

We have reworked the conclusions as requested with the evidence about the optimal dosing of supplementation. efficacy and safety were not two focal points in the text, so they were not integrated.

Abbreviations should be defined just once in the text to avoid repetition. For example, the term "CKD" was defined at least 18 times throughout the text.

Thank you for the suggestion. We have made the change as indicated.

While the paper briefly mentions potential interactions between nutrient status and other aspects of CKD management, such as pharmacotherapy and dietary interventions, exploration of these interactions and their implications for patient outcomes would strengthen the paper.

We have deepened these sections.

Ensure that Table 1 is referenced in the text.

Table 1 is now referenced in the text.

Round 2

Reviewer 1 Report

Comments and Suggestions for Authors

Following the recommendations, your manuscript clearly has been improved. Only one suggestion, in the final draft, please insert Table 1 before the Discussion section, rather than at the end of the article.

Reviewer 2 Report

Comments and Suggestions for Authors

The authors have effectively addressed all omissions.

Thank you for your thorough revisions. I have no further comments to add.